# NorSand4AI: A Comprehensive Triaxial Test Simulation Database for NorSand Constitutive Model Materials

Luan Carlos de Sena Monteiro Ozelim[1], Michéle Dal Toé Casagrande[1], and André Luís Brasil Cavalcante[1]

[1]Department of Civil and Environmental Engineering, University of Brasilia,Campus Universitário Darcy Ribeiro, SG12, Asa Norte. 70910-900, Brasilia-DF, Brazil

**Correspondence:** Luan Carlos de Sena Monteiro Ozelim (luanoz@gmail.com, ozelim@unb.br)

**Abstract.** To learn, humans observe and experience the world, collect data, and establish patterns through repetition. In scientific discovery, these patterns and relationships are expressed as laws and equations, data as properties and variables, and observations as events. In soil sciences, parametric models known as constitutive models are used to represent the behavior of natural and artificial materials. Due to its limitations for modeling real sands, the Modified Cam Clay model has been replaced
by the NorSand model in some situations where sand-like materials need to be modelled, especially when liquefaction may occur. For example, when the stacking of filtered tailings is considered, the height and speed of stacking are conditions that can create regions susceptible to liquefaction within the pile. The presence of a liquefaction trigger, especially in an undrained loading regime, can lead to the collapse of the structure. In cases of highly complex phenomena, such as liquefaction, data-driven modelling techniques can provide an impartial approach to learning using raw data from actual or simulated observations.
Creating data-driven constitutive models using deep learning techniques requires large and consistent datasets, which are challenging to acquire through experiments. Synthetic data can be generated using a theoretical function, but there is a lack of literature on high-volume and robust datasets of this kind. Digital soil models can be utilized to conduct numerical simulations that produce synthetic results of triaxial tests, which are regarded as the preferred tests for assessing soil's constitutive behavior. Therefore, for a material following the NorSand model, the present paper presents a first-of-its-kind database that addresses
the size and complexity issues of creating synthetic datasets for nonlinear constitutive modeling of soils by simulating both drained and undrained triaxial tests. Two datasets are provided: the first one considers a nested Latin Hypercube Sampling of input parameters encompassing 2000 soil types, each subjected to 40 initial test configurations, resulting in a total of 160000 triaxial test results. The second one considers nested quasi-Monte Carlos sampling techniques (Sobol and Halton) of input parameters encompassing 2048 soil types, each subjected to 42 initial test configurations, resulting in a total of 172032 triaxial
test results. Each simulation dataset comprises a $4000 \times 10$ matrix that can be used for general multivariate forecasting benchmarks, in addition to direct geotechnical and soil science applications. By using the quasi-Monte Carlo dataset and 49 of its subsamples, it is shown that the dataset of 2000 soil types and 40 initial test configurations is sufficient to represent the general behavior of the NorSand model. As a secondary outcome, this work introduces a Python script that links the established VBA implementation of NorSand to the Python environment. This enables researchers to leverage the comprehensive capabilities of
Python packages in their analyses related to this constitutive model.

# 1 Introduction

Metallic minerals are found mixed with various elements within rocks and can be extracted through mechanical and chemical processes conducted in mining plants. Elements without commercial value typically constitute the largest percentage of rock constituents, therefore, when removing the minerals of interest, large quantities of residual material are generated, considered as mining waste (tailings). On the other hand, materials excavated or generated during extraction activities (or mining) and during mine stripping, which have no economic value, are termed sterile and are usually disposed of in piles.

Furthermore, due to the high demand for metallic ore resources, rocks with reduced metal content have become economically viable for extraction, resulting in an increase in the quantity of tailings and necessitating larger disposal piles for this material. In the same vein, the volume of sterile material also increases, coupled with diminishing space at the mineral extraction site. Consequently, there arises a need to understand not only the optimal deposition methods for both tailings and sterile materials but also how these two substantial volumes of materials vie for available locations around the mine's operational center.

To ensure the safe maintenance, monitoring, and operation of tailings dams, it is crucial to conduct a comprehensive, impartial, and thoughtful evaluation of these structures during feasibility analysis, design, construction, operation, and decommissioning. This aims to mitigate geotechnical risks associated with the subject. Similarly, when examining the stacking of filtered tailings, related geotechnical risks emerge. In particular, the geotechnical risks of liquefaction in this type of disposal are significant and need to be assessed through appropriate constitutive modeling. In this case, the height and speed of stacking are conditions that can create regions susceptible to liquefaction within the pile. The presence of a liquefaction trigger, especially in an undrained loading regime, can lead to the collapse of the structure.

In this scenario, the NorSand constitutive model emerges as a suitable alternative to liquefaction modelling due to its relatively simple critical state formulation and low number of input parameters. This model is a generalized critical state model based on the state parameter $\psi$, as defined by Jefferies (1993):

$$\psi = e - e_c \tag{1}$$

where $e$ is the current void ration and $e_c$ is the void ratio at the critical state. The NorSand model emulates natural soil behavior by incorporating associated plasticity and limited hardening, which enables dilation similar to that observed in real soils. This limited hardening causes yielding during unloading conditions and provides second-order detail in replicating observed soil behavior (Silva et al., 2022; Jefferies and Been, 2015).

As emphasized by Silva et al. (2022), the significance and potential impact of the failure (liquefaction) of tailing dams/piles, especially within the scope of mine operation, imply that their geotechnical design cannot be confined to current practices of constitutive modeling. Jefferies and Been (2015) argue that the time and effort required to create models tailored to a specific project present limitations to the use of more comprehensive numerical analyses in engineering practice. This is because the cost of developing customized computational tools (such as implementation numerical solvers or refined constitutive models) can quickly exhaust the project's available budget. Therefore, it is important to find ways to create or modify models that can

accommodate the unique characteristics of the materials of interest without necessitating elaborate computational implementations.

Despite its suitability as a good modelling framework to assess static liquefaction (Sternik, 2015), the NorSand model still is bused upon premises which may not perfectly represent the behavior of all soil types. It is precisely in this context that the creation of data-driven and physically-informed metamodels emerges. These metamodels, when based on artificial intelligence techniques, especially machine learning (ML) and deep learning (DL), may be able to provide accurate and computationally cheap models, allowing them to be a perfect link between complex computational models and real-time data collection and monitoring. Such methods need to be trained on large-scale datasets and this is where the NorSand model comes handy: by using NorSand simulations as the training dataset, data-driven constitutive metamodels can then be fine-tuned using real test results. These models will combine the power of NorSand with the flexibility provided by data-driven approaches, enhancing the modelling capabilities for liquefaction.

Also, only recently the NorSand method has been implemented in commercial Finite Element softwares (Rocscience, 2022; Itasca Consulting Group, 2023; Bentley, 2022). Besides, regarding open-source distributions, only the Visual Basic (VBA) implementation presented by Jefferies and Been (2015) is available. Thus, another open-source implementation easily integrated into ML and DL modelling pipelines is desirable.

Thus, the current paper aims to address three main issues: the quantity and complexity of synthetic datasets for nonlinear constitutive modeling of soils and the availability of open-source implementations of the NorSand constitutive model. The first two aspects are addressed by simulating both drained and undrained triaxial tests. Two datasets are provided: the first one considers a nested Latin Hypercube Sampling of input parameters encompassing 2000 soil types, each subjected to 40 initial test configurations, resulting in a total of 160000 triaxial test results. The second one considers a nested quasi-Monte Carlos sampling (Sobol and Halton) of input parameters encompassing 2048 soil types, each subjected to 42 initial test configurations, resulting in a total of 172032 triaxial test results. Each simulation dataset comprises a $4000 \times 10$ matrix that can be used for general multivariate forecasting benchmarks, in addition to direct geotechnical and soil science applications. By using the quasi-Monte Carlo dataset and 49 of its subsamples, it is shown that the dataset of 2000 soil types and 40 initial test configurations is sufficient to represent the general behavior of the NorSand model. Then, the third aspect is considered by presenting an implementation which connects the well-known VBA implementation to the Python environment. We will use the VBA code as the "processing kernel" of our Python implementation, taking advantage of the years of tests and validation of the algorithm provided by Jefferies and Been (2015). This new Python code allows other researchers to use the full power of Python packages during their analyses involving NorSand.

The paper is structured as follows: Section 2 presents the general concepts of data-driven metamodels, with special emphasis to soil constitutive modelling. Then, Section 3 introduces the Norsand model. Section 4, on the other hand, presents the Methods considered in the present paper. Then, Section 5 describes the Data Records associated with the present paper, while Section 6 presents the Technical Validation of the results. Section 7 presents some Usage Notes and Codes considered in the present paper. Finally, Section 8 presents the conclusions.

## 2 Data-driven metamodels

Montáns et al. (2019) emphasize that human learning involves observing and experiencing the world, collecting data, and identifying patterns through repeated experiments. Scientific discovery involves formalizing these patterns and relationships into laws and equations, transforming data into properties and variables, and converting observations into events. Although laws and equations aid learning, the classical learning process in science is often slow and expensive, requiring extensive observation and experimentation to understand the main variables and their impact on the phenomenon. Data-driven procedures, on the other hand, seek, if possible, an implicitly unbiased approach to our learning experience based on raw data from actual or synthetic observations. These procedures have the added advantage of testing correlations between different variables and observations, learning unanticipated patterns in nature, and allowing us to discover new scientific laws or even make predictions without the availability of such laws.

The recent rapid increase in the availability of measurement data from physical systems as well as from massive numerical simulations has stimulated the development of many data-driven methods for modeling and predicting dynamics. At the forefront of data-driven methods are Deep Neural Networks (DNNs). DNNs not only achieve superior performance for tasks such as image classification, but have also proven effective for future state prediction of dynamical systems (Haghighat et al., 2021). A key limitation of DNNs, and similar data-based methods, is the lack of interpretability of the resulting model: they are focused on prediction and do not provide governing equations or clearly interpretable models in terms of the original set of variables. An alternative data-based approach uses symbolic regression to directly identify the structure of a nonlinear dynamical system from data (Schmidt and Lipson, 2009). This works remarkably well for discovering interpretable physical models, but symbolic regression is computationally expensive and can be difficult to scale to large problems (Montáns et al., 2019).

### 2.1 Data-driven constitutive modelling

In order to create metamodels from Neural Networks (NN), this type of approach generally requires a priori calibration of the algorithms from data considered to be representative of material behavior (He et al., 2021). For example, NNs have been applied to model a variety of materials, including concrete materials (Ghaboussi et al., 1991), hyperelastic materials (Shen et al., 2005), viscoplastic steel material (Furukawa and Yagawa, 1998), and homogenized properties of mixed structures (Lefik and Schrefler, 2003). Once calibrated, NN-based constitutive models have been integrated into finite element codes to predict path- or rate-dependent material behaviors (Lefik and Schrefler, 2003; Hashash et al., 2004; Jung and Ghaboussi, 2006; Stoffel et al., 2019).

Recently, DNNs with special mechanistic architectures, such as Recurrent Neural Networks (RNNs), have been applied to path-dependent materials (Wang and Sun, 2018; Mozaffar et al., 2019; Heider et al., 2020). It is clear that this type of approach has found significant applications in a wide range of engineering fields, as reinforced by He et al. (2021), when they argue that data-driven computation with physical constraints is an emerging computational paradigm that allows the simulation of complex materials directly based on the materials database and disregards the classical constitutive model construction.

To develop a data-driven constitutive model, a substantial and reliable dataset is necessary. However, obtaining a sufficiently large dataset for soil science can be challenging since experimental data is often limited and inadequate for training ML and DL algorithms. Generating synthetic data using a theoretical function can be a useful alternative, as it allows for the creation of an unlimited supply of data (Zhang et al., 2021a).

The literature suggests that data-driven models should initially be developed using synthetic datasets to establish a general framework, which can later be applied to experimental datasets to enhance the model's robustness and aid in discovering potential mechanisms of soil behavior (Zhang et al., 2021a). By calibrating constitutive models on synthetic datasets, the impact of experimental and measurement errors on the mapping ability of machine learning algorithms can be eliminated (Zhang et al., 2020). Therefore, creating large and reliable synthetic datasets is a crucial step in constructing data-driven constitutive models.

### 2.1.1 Data-driven soil constitutive models

Currently, there is a lack of robust and high-volume datasets in the literature for soil modeling tasks. One effective method to generate synthetic datasets is through numerical simulations performed on digital soil models. Typically, these simulations involve selecting a parametric constitutive model, sampling some parameters, and running simulations that mimic real-world test setups. In soil modeling, triaxial tests are commonly simulated using conventional physics-driven constitutive models, such as simple monotonic Konder's expression (Basheer, 2000) or more advanced models like the Modified Cam Clay (MCC) (Fu et al., 2007; Zhang et al., 2023).

In particular, a simple sand shear constitutive model was used to generate synthetic datasets in the work of Zhang et al. (2021b). A total of fourteen curves were generated to develop the ML-based constitutive model (nine curves for training and five curves for testing).

On the other hand, the MCC constitutive model was utilized to produce a benchmark stress–strain dataset of a virtual soil in the work of Zhang et al. (2023). In that study, a total of 250 soil types were considered, with 125 being part of the training dataset and the remaining 125 in the testing dataset. Considering all the initial states in the paper by Zhang et al. (2023), 1125 sets of stress–strain samples were employed as the training dataset, while 1250 sets of stress–strain samples constituted the testing dataset.

The MCC model has been a fundamental element in numerous complex models developed in recent times (Yao et al., 2008). However, this model and its variations are not well-suited for depicting the behavior of actual sands due to their insufficient representation of key features such as yielding and dilation. This is because these models assume that soils denser than the critical state line are over-consolidated, resulting in an unrealistically high stiffness and excessively exaggerated strength (Woudstra, 2021). As indicated in the Introduction section, the NorSand constitutive model presents clear advantages over MCC model and, therefore, shall be described in detail in the next Section.

## 3   NorSand

The NorSand constitutive model is a comprehensive critical state model that effectively accounts for the impact of void ratio on soil behavior, providing a robust framework for modeling static liquefaction in engineering applications. A distinctive characteristic of soils is that their void ratios or relative densities influence their mechanical properties. In this regard, NorSand, as a constitutive model, aptly elucidates changes in soil behavior resulting from variations in void ratio (Jefferies and Been, 2015).

Within the Critical State Soil Mechanics (CSSM) framework, NorSand aligns with widely used models like Modified Cam Clay (Roscoe and Burland, 1968). CSSM is founded on two principles: 1) the presence of a unique failure locus known as the Critical State Locus (CSL), and 2) the assertion that shear strain guides soil toward the CSL.

The primary limitation of Modified Cam Clay, especially when applied to sands, lies in its inability to capture the dilation behavior observed in dense sands. Moreover, it proves inadequate in predicting the behavior of loose sands and is unsuitable for addressing liquefaction-related issues. NorSand's key advantage lies in its incorporation of a state parameter, representing the difference between the current void ratio of the soil and its critical state. This approach uniquely relates soil dilation or compaction to the state parameter (Rocscience, 2022).

NorSand stands out for its ease of use, particularly for practical geotechnical engineers. It relies on a minimal set of material properties, conveniently measurable through standard laboratory tests. The model effectively captures a wide range of soil behaviors influenced by varying density and confining stress. The key additional parameter, beyond what is necessary for defining a Modified Cam Clay model, is the state parameter. In situations where precision in representing volume change is crucial, the added effort required for parameter determination is more than justified.

Developed initially for sands based on observations in large-scale hydraulic fills such as tailing dams, NorSand applicability extends beyond, encompassing any soil where particle-to-particle interactions are controlled by contact forces and slips, rather than cohesive bonds. Present applications of NorSand span a range from well-graded tills to sands and clayey silts (Jefferies and Been, 2015).

The input parameters of the NorSand model are presented in Table 1. The sampling ranges adopted come from literature results on the behavior of real granular materials. An initial version of such ranges was first presented by Jefferies and Shuttle (2002) and has been updated ever since. The ranges presented in Table 1 reflect the latest compilation available and reported by Jefferies and Been (2015). This way, practitioners will especially benefit from the datasets generated, since the parameters involved have been chosen as to represent real granular materials. Table 1 also present the meaning of each parameter in the column "Description".

**Table 1.** Input values for NorSand model also used as inputs for the NorSandTXL VBA routine (Jefferies and Been, 2015).

| Parameter Class | Parameter | Sampling range | Units | Description |
|---|---|---|---|---|
| | | Soil properties | | |
| CSL parameters | $\Gamma|_{p'=1kPa}$ | [0.9,1.4] | - | CSL mean effective stress at $p' = 1kPa$ |
| | $\lambda$ | [0.01,0.07] | $(\ln \text{kPa})^{-1}$ | Slope of CSL defined on base $e$ |
| Plasticity | $M_{tc}$ | [1.2,1.5] | - | Critical friction ratio, with triaxial compression as a reference condition |
| | $N$ | [0.2,0.5] | - | Volumetric coupling parameter |
| | $\chi_{tc}$ | [2,5] | - | Relates minimum dilatancy to corresponding $\psi$, with triaxial as a reference condition |
| | $H_0$ | [75,500] | - | $H$ is the loading plastic hardening modulus, such that: |
| | $H_\psi$ | [200,500] | - | $H = H_0 + H_\psi \psi$ |
| Elasticity | $G_{max}|_{p'_0}$ | [30,100] | *MPa* | Shear modulus at $p' = p'_0$ |
| | $G_{exp}$ | [0.1,0.6] | - | Exponent of nonlinear shear modulus change with stress, $G_{max} = G_{max}|_{p'_0} (p'/p'_0)^{G_{exp}}$ |
| | $\nu$ | [0.1,0.3] | - | Poisson's ratio |

| Parameter Class | Parameter | Sampling range | Units | Description |
|---|---|---|---|---|
| | | Initial Soil State | | |
| Stress and Deformability | $\psi_0$ | [-0.2,$\psi_{max}/5$] | - | Initial critical state parameter, where $\psi_{max} = M_{tc}/(\chi(1+N))$ |
| | $p'_0$ | [50,1000] | kPa | Initial mean effective stress |
| | $K_0$ | [0.8,1.2] | - | Geostatic stress ratio |
| | OCR ("$R$") | [0.5,3] | - | Overconsolidation ratio |

## 4 Methods

### 4.1 Data Generation

The NorSandTXL program is an Excel spreadsheet with all coding in the VBA environment and can be downloaded at http://www.crcpress.com/product/isbn/9781482213683, as indicated in the book by Jefferies and Been (2015). This particular spreadsheet simulates drained and undrained triaxial tests of materials governed by the NorSand constitutive model. The input features available in NorSandTXL were presented in Table 1.

In order to massively simulate triaxial test conditions for materials following the NorSand constitutive model, a Python
routine has been developed. This routine performs two main steps: sampling and simulation. For the sampling process, all 14 input parameters are sampled in a nested manner, as there are two levels of hierarchy in the parameters: the higher level deals with the soil properties, which are unique for a given material, while the lower level considers the initial soil state during the

triaxial tests. As a result, the sampling process needs to: a) account for different types of materials and b) for each type of material, consider several testing conditions.

Thus, the following sampling procedure is considered to account for $n_{soils}$ types of soils under $n_{conditions}$ initial testing conditions:

- Sample the soil properties (the first ten parameters in Table 1), obtaining a vector of properties $sp_i$, $i = 1, ..., n_{soils}$, such that $sp_i \in \mathbb{R}^{10}$. The sampling is performed using the centered Latin hypercube sampling algorithm implemented in the *chaospy* package (Feinberg and Langtangen, 2015) with a maximin criterion.

- For each $sp_i$, the initial testing conditions (the last four parameters in Table 1) are sampled using the standard Latin hypercube sampling algorithm implemented in the *chaospy* package (Feinberg and Langtangen, 2015) with a ratio criterion. This way, the vectors $ic_{i,j} \in \mathbb{R}^4$, $j = 1, ..., n_{conditions}$ are obtained for each $sp_i$. The maximum value of $\psi_0$ is set to $\psi_{max}/5$ (as indicated in Table 1) for numerical stability. Additionally, to make the $ic_{i,j}$ different for each $sp_i$, the random seed of the sampling algorithm is changed for each $i$.

From the procedure above, the matrix $In$ of input parameters is obtained, whose rows are NorSandTXL input vectors obtained by concatenating each $sp_i$ with all the $ic_{i,j}$, i.e., $[concat(sp_1, ic_{1,1}), concat(sp_1, ic_{1,2}), ..., concat(sp_{n_{soils}}, ic_{n_{soils}, n_{conditions}})]$, where $concat$ denotes a concatenation operation between vectors. This implies that $In$ is a $(n_{soils} n_{conditions})$ by 14 matrix.

The simulation step, on the other hand, involves opening the Excel spreadsheet provided in the book by Jefferies and Been (2015), inputting the sampled parameters, running both drained and undrained simulations for the input parameters and collecting their respective results, finally saving them in *.h5* format files for posterior processing.

The file extension *.h5* is associated with the Hierarchical Data Format (HDF5) (The HDF Group, 1997-2023), which is a type of high-performance distributed file system. It is specifically designed to manage large and complex data sets efficiently and flexibly. Additionally, it enables a self-describing file format that is portable and supports parallel I/O for data compression (Lee et al., 2022), and has shown superior performance with high-dimensional and highly structured data (Nti-Addae et al., 2019). Literature indicates that the HDF5 has been popular in scientific communities since the late 1990s (Lee et al., 2022), which is evident by the large number of open-source and commercial software packages for data visualization and analysis that can read and write HDF5 (Group, Accessed on April 24, 2023). As a result, this is the data format chosen for the present paper.

## 4.2 Sample size validation

The samples generated using the methods in the last subsection need to be sufficiently large in order to represent the general behavior of the NorSand model. The best way to show that the sample size is sufficient is to study how a model calibrated (or trained) on a given dataset performs. So, we chose the most direct (and actually most important) learning task one could face while working with the datasets generated: back-calculation of the constitutive parameters of the model based solely on the

triaxial test results. In short, from the triaxial tests we will learn the values of the parameters which govern the behavior of the material.

This way, it is possible to recall that a total of 14 parameters (10 constitutive and 4 related to test conditions) are used to generate the triaxial test results ($4000 \times 10$ array where 4000 denotes the number of time steps of the loading process and 10 is the number of quantities monitored during the test). From last subsection's notation, Let $In_i$ (shape 1x14) be the $i$-th row of the

$In$ matrix, which contains the constitutive parameters, and let $ttu_i$ and $ttd_i$ be the results of the triaxial test under undrained and drained conditions, respectively (4000x10 arrays, each) obtained by using these parameters on the NorSandTXL routine.

We will consider the following learning problem: From a sample of input parameters $In = In_{n,m}$, which considers $n$ different types of soil and $m$ different test configuration (therefore with $nm$ rows), we will use the $ttu_i$ (or $ttd_i$), for $i = 1, ..., nm$, to learn the vectors of parameters $In_i$, for $i = 1, ..., nm$. We wish to investigate what are the values of $n$ and $m$ that

suffice to produce an accurate representation of the model. In order to do so, following standard learning tasks in a Machine Learning context, we need training, validation and testing data. It is worth noticing that our methodology needs to be robust, so we indeed need the validation dataset because hyperparameter tuning will be performed.

The dataset obtained by following the methods of the first subsection was generated by a Latin Hypercube Sampling (LHS) algorithm, which is known to provide low-discrepancy sequences of values (i.e., the samples are spread in the domain of the

sampled variables). Despite being a really powerful technique, LHS does not have an interesting property: sequences obtained by LHS are not extensible. To put it simply, being extensible means that a sample of size $j$ contains the values of the sample of size $k$, $j > k$. This way, it would not be possible to sub-sample from our original sample $In$ in order to build smaller datasets without loosing the space-filling capability of the dataset. This way, we needed to consider another sampling scheme to perform our investigation.

We chose to combine two quasi-Monte Carlo low discrepancy sequence generation techniques (Sobol (Sobol', 1967) and Halton (Halton, 1960)), which are also extensible, to perform our tests. In that case, we generated a dataset with $n = 2048$ and $m = 42$ using Sobol sampling for the constitutive parameters (10 parameters) and Halton sampling for the experimental test condition variables (4 variables) using the SciPy Python package (Virtanen et al., 2020). Both sequences have been scrambled (Owen and Rudolf, 2021) to improve their robustness for space filling. By using these parameters, we ran the NorSandTXL

routine in the same manner as described in the first subsection and obtained the corresponding triaxial test results for both drained and undrained cases. Let us call this new dataset and $qIn_{2048,42}$.

By using the extensibility property of the sequences considered, 49 sub-samples were taken: $qIn_{n,m}$ for $n$ in [32, 64, 128, 256, 512, 1024, 2048] and $m$ in [6, 12, 18, 24, 30, 36, 42]. One may see that powers of 2 were used as sample sizes for the Sobol sampling scheme, which is standard and derives from its implementation in *scipy.stats*. It is worth noticing that, in general,

none of the entries of $In_{n,m}$ will be in $qIn_{n,m}$, which indicates that using $qIn_{n,m}$ for training and validation and $In_{n,m}$ for testing does not allow for any data "leakage". Besides, there is a clear benefit in using $In_{n,m}$ as a test set: all the models will be tested on the same dataset.

For the learning task considered, we used the *scikit-learn* Python package (Pedregosa et al., 2011) and chose 4 algorithms: Ridge Regressor, KNeighbors Regressor and two variants of the Ridge Regressor which incorporate nonlinear mappings of

the input and output values. The first two algorithms mentioned belong to two different classes: linear and neighbors-based regressors. They were chosen to illustrate how different types of algorithms learn our chosen task. The variants of the Ridge Regressor were chosen to account for nonlinearities by using the kernel trick. Considering the high dimensionality of the input datasets, using traditional kernels is not computationally feasible, so we used Nystroem kernels (Yang et al., 2012), which approximate a kernel map using a subset of the training data. By combining Nystroem kernels and Ridge Regressors, we can

map the inputs to a nonlinear feature space and then consider a linear regression on these features. This is a similar approach as the one considered to build Support Vector Machine Regressors, but with a slightly different regularization for the decision boundary.

      We also considered mapping the output values (14 parameters, in our case) to the [0,1] range by combining the *scikit-learn* implementations of TransformedTargetRegressor and QuantileTransformer, which transforms the target values (outputs of the

pipeline) to follow a uniform distribution. Therefore, for a given component, this transformation tends to spread out the most frequent values. It also reduces the impact of (marginal) outliers (Pedregosa et al., 2011). For all the algorithms considered, we also used a QuantileTransformer to preprocess the input values.

      This way, Figure 1 presents the methodology proposed and applied to assess the quality of the sample size. In the present paper, the LHS-generated dataset with $n_{soils} = 2000$ and $n_{conditions} = 40$, whose input parameter matrix is $In_{2000,40}$, will

have its sufficiency assessed.

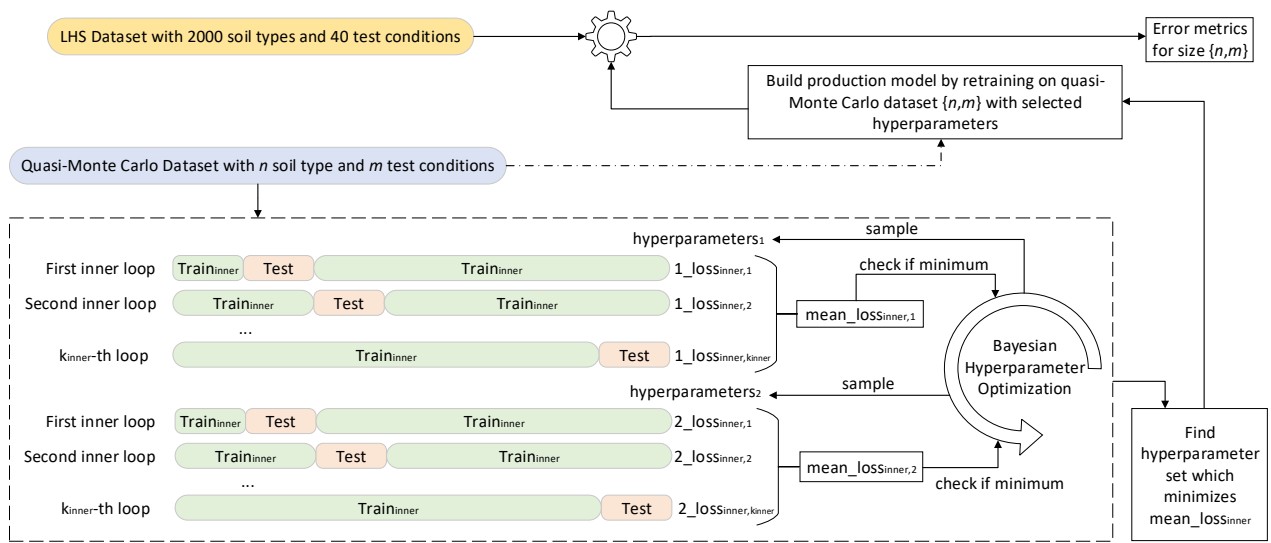

**Figure 1.** Methodology used to assess the sufficiency of the dataset containing 2000 soil types and 40 test conditions to represent the general behavior of the NorSand model

      It is possible to describe the workflow in Figure 1 as:

      For $n$ in [32,64,128,256,512,1024,2048]:

For $m$ in [6,12,18,24,30,36,42]:

- For each simulated triaxial test corresponding to the parameters matrix $qIn_{n,m}$, select only the columns corresponding to $\epsilon_1$, $p'$, $q$ and $e$ (axial strain, mean effective stress, deviatoric stress and void ratio, respectively), which are the variables commonly measured and reported. The other 7 columns are manipulations of these three. This reduced simulation dataset is of shape 4000x4.

- Each triaxial test simulation may have different start/end values for $\epsilon_1$, so it is important to "align" all the test considered. By alignment we mean that all the tests will have measurements for the same values of $\epsilon_1$. This will enable us to use this variable as an index and, therefore, decrease the dimensionality of each triaxial test simulation from 4000x4 to 4000x3.

- Downsample the 4000 timesteps to 40, by using evenly spaced values on a logarithmic scale (function *logspace* from Python package *numpy*: more values in the beginning of the time steps, where more changes are observed). This reduces each simulated triaxial test corresponding to the parameters matrix $qIn_{n,m}$ from 4000x10 to 40x3. The concatenation of all triaxial test results corresponding to the parameters matrix $qIn_{n,m}$ shall be named $qInN_{n,m}$ and is of size $(nm, 40, 3)$.

- Perform a GroupKFold cross-validation scheme to find the best hyperparameters of an algorithm $A$ using $qInN_{n,m}$ and inputs and $qIn_{n,m}$ as outputs. The loss function considered during the GroupKFold cross-validation is the mean absolute percentage error across all folds;

- Retrain the algorithm $A$ using all $qInN_{n,m}$ and $qIn_{n,m}$ after fixing the hyperparameters as the optimal ones obtained during the cross-validation scheme;

- Test the trained algorithm $A_t$ on $In_{n_h,m_h}$, where $n_h$ and $m_h$ are the hypothesized sufficient number of materials and test conditions, respectively;

- Obtain the mean absolute percentage error in the predictions of all the 14 input parameters corresponding to $In_{n_h,m_h}$;

- Get the overall mean error, corresponding to all the input parameters.

As described, for training and validation, we considered a GroupKFold cross validation technique, which is a K-fold iterator variant with non-overlapping groups (Pedregosa et al., 2011). This approach makes sure no material (group) is present both in train and validation set, which would lead to data "leakage".

A Bayesian optimization was performed to look for the best hyperparameters using the cross-validation folds generated. This process was carried out using the *Hyperopt* Python package (Bergstra et al., 2015), which considers Tree-structured Parzen Estimators. The search space for the Ridge and KNeighbors Regressors are the ones considered in the *Hyperopt-Sklearn* Python package (Komer et al., 2014). For the Nystroem kernel, a custom search space was defined and consisted of: 'gamma' parameter uniformly on [0,1]; 'n_components' parameter as a random equi-probable choice among [600,1200,1800]; 'kernel' parameter as a random equi-probable choice among ["additive_chi2", "chi2", "cosine", "linear", "poly", "polynomial",

"rbf","laplacian", "sigmoid"]; 'degree' parameter as the integer value truncation of an uniform random variable on [1, 10] and 'coef0' parameter uniformly on [0,1].

Finally, after the best hyperparameters are found, they are fixed and the algorithm $A$ is retrained with the full dataset $qInN_{n,m}$. This calibrated version is then used to test the quality of the model on the triaxial test results corresponding to the dataset $In_{n_h,m_h}$. Then, the errors obtained for each model are plotted and analyzed. The reader may find the complete codes used to implement the steps above in (Ozelim et al., 2023b).

## 5    Data Records

In the present paper, it is shown that the LHS-generated dataset with $n_{soils} = 2000$ and $n_{conditions} = 40$ is a sufficient dataset. Thus, the folder containing such dataset can be found in Ozelim et al. (2023a) and has the following structure:

NorSandTXL_H5 \Simus\**TT**\Par_**X**_**Y**.h5

where **TT** stands for the test type (Drained or Undrained), **X** is the material index (from 0 to 1999) and **Y** is the sequential index for the input parameters (from 0 to 79999).

Each *Par_X_Y.h5* file contains a dataset titled 'NorSandTXL' which includes the simulation results. By design, the Nor-SandTXL Excel spreadsheet considers 4000 strain steps to go from zero to approximately 20% nominal axial strain at the end of the simulated test. The authors of the spreadsheet indicate that this amount is both convenient and sufficient (Jefferies and Been, 2015). On the other hand, for a triaxial effective stress state with vertical stress $\sigma'_a$ (kPa) and confining stress $\sigma'_r$ (kPa), a total of 10 entities are reported from the tests, which are: $\epsilon_1$ (axial strain); $\epsilon_v$ (volumetric strain); $p' = (\sigma'_a + 2\sigma'_r)/3$ (mean

effective stress in kPa); $q = \sigma'_a - \sigma'_r$ (deviatoric stress in kPa); $e$ (void ratio); $p_i/p'$ (stress ratio); $(p_i/p')_{max}$ (maximum stress ratio); $\psi$ (state parameter); $D_p$ (dilation) and $\eta = q/p'$. Thus, the dataset is a $4000 \times 10$ array, as presented in Table 2.

**Table 2.** The 'NorSandTXL' dataset present in each *Par_X_Y.h5* file.

| $\epsilon_1$ | $\epsilon_v$ | $p'$ | $q$ | $e$ | $p_i/p'$ | $(p_i/p')_{max}$ | $\psi$ | $D_p$ | $\eta$ |
|---|---|---|---|---|---|---|---|---|---|
| 0 | 0 | 200 | 0 | 0.9021 | 0.42306 | 1 | 0 | 0.92603 | 0 |
| 0.06097 | 0.04314 | 209.561 | 28.2795 | 0.90128 | 0.40376 | 1 | 0 | 0.92603 | 0.13495 |
| 0.07544 | 0.05481 | 210.703 | 31.7059 | 0.90106 | 0.40811 | 0.99319 | 0.001981083 | 1.31505 | 0.15048 |
| 0.0897 | 0.06628 | 211.821 | 35.0611 | 0.90084 | 0.41236 | 0.99284 | 0.002085266 | 1.29952 | 0.16552 |
| | | | | ... | | | | | |
| 19.3293 | 2.10004 | 387.564 | 562.29 | 0.86216 | 1.00101 | 1.00087 | -0.000251146 | -0.00146 | 1.45083 |
| 19.3334 | 2.10003 | 387.564 | 562.29 | 0.86216 | 1.00101 | 1.00087 | -0.000251018 | -0.00146 | 1.45083 |
| 19.3374 | 2.10002 | 387.564 | 562.29 | 0.86216 | 1.00101 | 1.00087 | -0.000250889 | -0.00146 | 1.45083 |

It is worth noticing that the values stored are of the type *float32*, which is sufficient for the applications envisioned for the dataset. In addition to the simulation results, the dataset also contains the attributes shown in Table 3. The correspondence

between the attributes, whose data type is either *float32* or *<U7* (fixed-length character string of 7 Unicode characters), and NorSandTXL input parameters is also presented in Table 3.

**Table 3.** Attributes of the 'NorSandTXL' dataset present in each *Par_X_Y.h5* file.

| Attribute | Parameter/Value |
|-----------|-----------------|
| 'Gamma' | $\Gamma\|_{p'=1kPa}$ |
| 'lambda' | $\lambda$ |
| 'Mtc' | $M_{tc}$ |
| 'N' | $N$ |
| 'Xtc' | $\chi_{tc}$ |
| 'H0' | $H_0$ |
| 'Hy' | $H_\psi$ |
| 'Gmax_p0' | $G_{max}\|_{p_0'}$ |
| 'G_exp' | $G_{exp}$ |
| 'n' | $\nu$ |
| 'Psi_0' | $\psi_0$ |
| 'p0' | $p_0'$ |
| 'K0' | $K_0$ |
| 'OCR' | OCR ("$R$") |
| 'Type' | Drained or Undrained |

It is easy to see that the dataset attributes in each file allow for a complete reproduction of the results, if desired. The units of the parameters are consistent with NorSandTXL, as presented in Table 1.

In order to prove the sufficiency of $In_{2000,40}$, we generated the dataset $qIn_{2048,42}$ following the methods previously presented. This latter dataset is also available at Ozelim et al. (2023a) with a similar folder structure. In that case, the upper-level folder is named $NorSand\_2048\_42$. It is worth noticing that, due to upload difficulties, $NorSand\_2048\_42$ was split as $NorSand\_2048\_42\_Drained$ and $NorSand\_2048\_42\_Undrained$, where each file contains the simulations for drained and undrained scenarios, respectively.

## 6 Technical Validation

Considering that the engine running the triaxial test simulations is the Excel spreadsheet presented in the book by Jefferies and Been (2015) and that such spreadsheet has been extensively validated by both academia and industry, there is no need to discuss the technical quality of the dataset. On the other hand, it is necessary to show that $In_{2000,40}$ suffices to cover the general behavior of the NorSand models.

By following the methods previously described and plotting the mean absolute percentage error (MAPE) result of the 49 models (each trained and validated with samples of different sizes subsampled from $qIn_{2048,42}$) Figure 2 and 3 were obtained

for drained and undrained conditions, respectively. The 4 algorithms considered were Ridge, KNeighbors, Ridge-K (with nonlinear kernel on inputs) and Ridge-KT (with nonlinear kernel on inputs and also QuantileTransformer on the outputs). It is clear in the figures that, for contours of 0.5% gains in MAPE, the sample size of 2000x40 is actually more than enough for the learning task considered. This can be stated by noticing that the contours with lower error encompass samples with an exponential range of sizes (the x-axis is in log scale). This indicates a really small gradient on the error in the $nxm$ space,

implying a good sample size. This happens for all 4 algorithms, indicating that not only linear and neighbors-based regressors have reached their maximum ability to learn, but also the nonlinear variants considered. It can be seen that the two nonlinear transformations applied (to inputs and to both inputs and outputs) present a similar behavior, although with considerably smaller MAPEs.

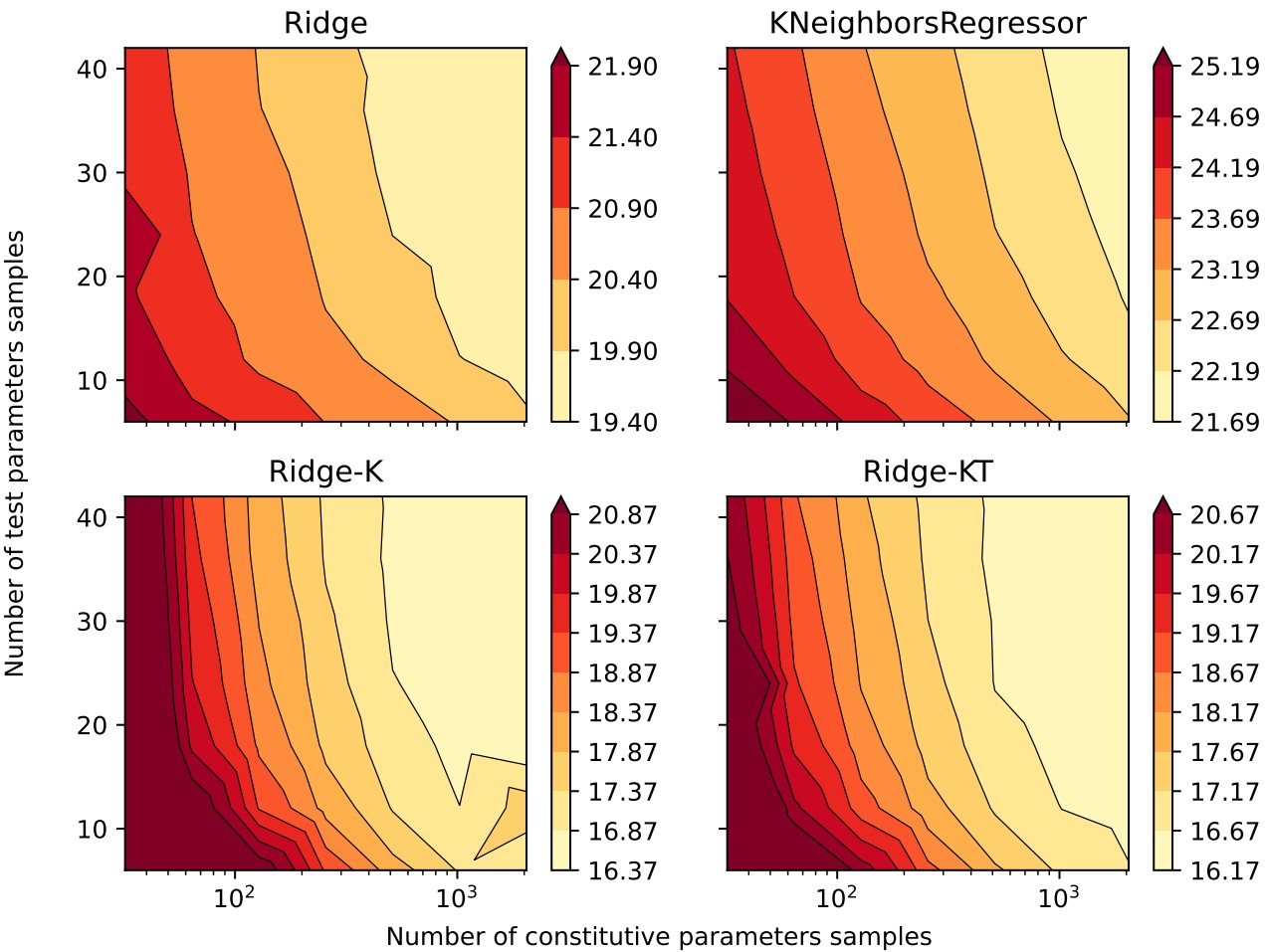

**Figure 2.** Mean absolute percentage error for all the 14 parameters after being back-calculated solely from drained triaxial test results.

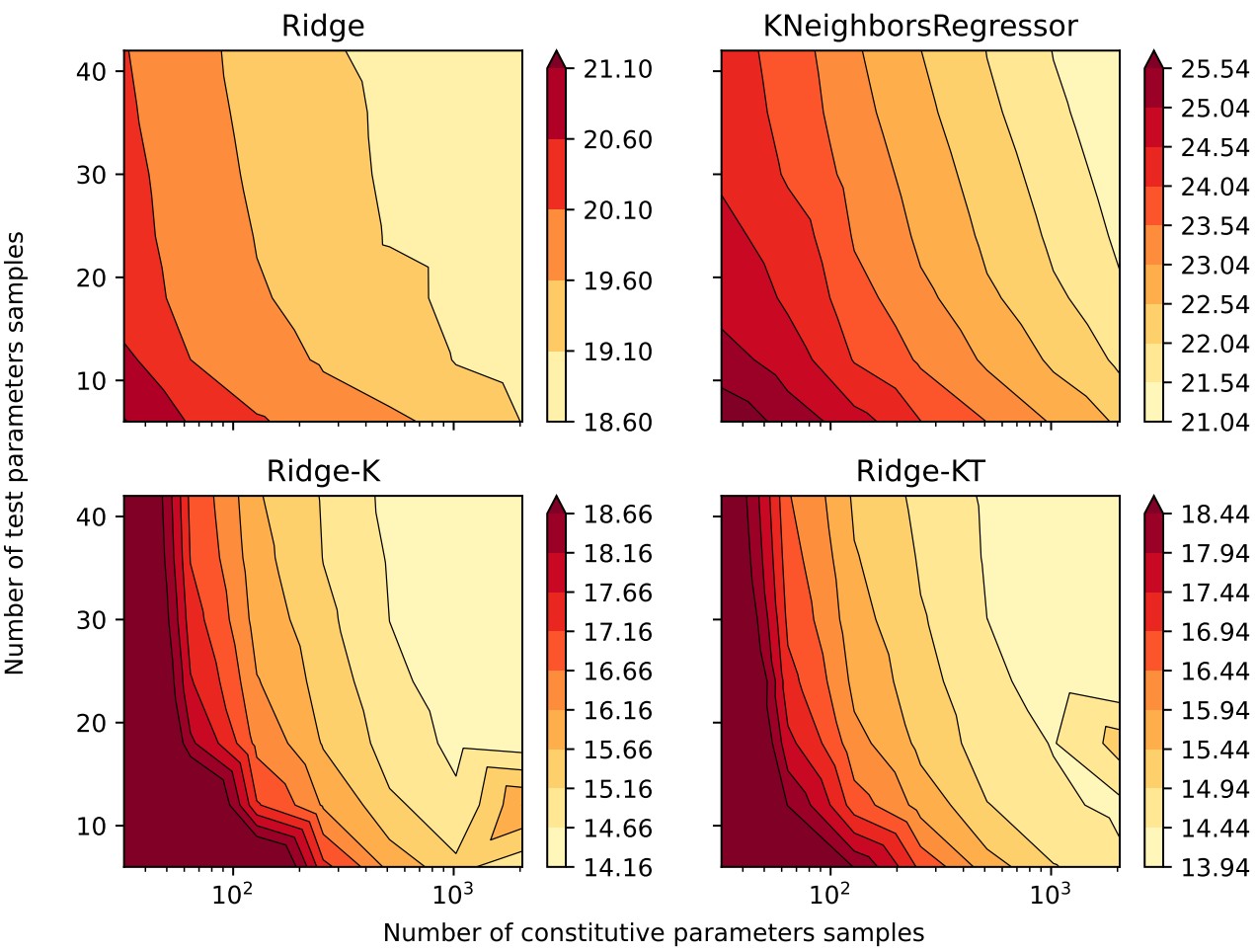

**Figure 3.** Mean absolute percentage error for all the 14 parameters after being back-calculated solely from undrained triaxial test results.

Due to the space-filling qualities of both $In_{2000,40}$ and $qIn_{2048,42}$, $qIn_{2048,42}$ can also be considered a sufficient dataset to
355 represent the NorSand model.

## 7 Usage Notes and Codes

In Python, the *h5py* package provides all the necessary tools to interact with the *.h5* files produced and made available in the NorSand4AI dataset. Depending on the intended application, it might be beneficial to down-sample the $4000 \times 10$ matrix to increase the axial strain increments. This can be accomplished using standard Python packages such as *pandas* and *numpy*. In
this section, the codes used to generate the datasets are presented. At first, the following Python packages need to be imported:

```
1: import numpy as np
```

```
 2: import math
 3: import pandas as pd
 4: import xlwings as xw
 5: import string
 6: from skopt.space import Space
 7: from skopt.sampler import Lhs
 8: from scipy.stats import qmc
 9: import os
10: import h5py
```

The packages *numpy*, *math* and *pandas* are required for data manipulation and numeric calculations. The *xlwings* package is needed to bridge Python and Excel. On the other hand, the *string* package is necessary to convert the (row-column) positional encoding to the (row-letter) alphanumeric encoding used in Excel. For the Latin Hypercube sampling procedure, *skopt* is required, while *qmc* from *scipy.stats* is need for the quasi-Monte Carlos sampling. Lastly, for creating folders and files, both *os* and *h5py* should be imported.

Let $dictpos$ be a dictionary that points to the locations in the spreadsheet of the cells corresponding to each input parameter. Additionally, let $dict\_ranges\_material$ and $dict\_ranges\_test$ be dictionaries specifying the sampling ranges of the input parameters. For this paper, these dictionaries are:

```
1: dictpos = {"Gamma":[6,4],"lambda":[7,4],"Mtc":[14,4], "N":[15,4],
2:        "Xtc": [16,4],"H0":[17,4],"Hy":[18,4], "Gmax_p0":[21,4],
3:        "G_exp": [22,4], "nu":[23,4],"Psi_0":[27,4],"p0":[29,4],
4:        "K0": [30,4], "OCR": [32,4]}
5: dict_ranges_material = {"Gamma":[0.9,1.4],"lambda":[0.01,0.07],"Mtc":[1.2,1.5],"N":[0.2,0.5],
6:        "Xtc": [2,5],"H0":[75,500],"Hy":[200,500], "Gmax_p0":[30,100],
7:        "G_exp": [0.1,0.6], "nu":[0.1,0.3]}
8: dict_ranges_test = {"Psi_0":[−0.2,0.2],"p0":[50,1000],
9:        "K0": [0.8,1.2], "OCR": [0.5,3]}
```

## 7.1  Simply run NorSand in Python

If one seeks to simply run NorSand in Python, the function $run\_NorSand$ can be used. Its inputs are:

- $final\_comp$: input parameters as a numpy array of shape (1,14). The parameters need to be inserted in the same order as $dictpos.keys()$, i.e., ['Gamma', 'lambda', 'Mtc', 'N', 'Xtc', 'H0', 'Hy', 'Gmax_p0', 'G_exp', 'nu', 'Psi_0', 'p0', 'K0', 'OCR'].

- $dictpos$: dictionary to locate the parameters inside the spreadsheet.

- $path\_root$: path of the spreadsheet "NorTxl.xlsm", obtained at http://www.crcpress.com/product/isbn/9781482213683

– $type\_v$: type of the simulation (either "Drained" or "Undrained")

```
1:  def run_NorSand(final_comp,dictpos,path_root,type_v):
2:      letters = list(string.ascii_uppercase)
3:      wb = xw.Book(path_root)
4:      app = wb.app
5:      macro_vba = app.macro("'NorTxl.xlsm'!RunSim")
6:      macro_vba_type = app.macro("'NorTxl.xlsm'!ChangeSimMode")
7:      ws = wb.sheets["Params_&_Plots"]
8:      results_comp = []
9:      for new_v in final_comp:
10:         for nv,ps in zip(new_v,dictpos.values()):
11:             pl,pc = ps
12:             pfinal = letters[pc−1]+str(pl)
13:             ws[pfinal].value = nv
14:         if ws["D34"].value == type_v:
15:             pass
16:         else:
17:             macro_vba_type()
18:         macro_vba()
19:         ws_results = wb.sheets["Txl_SimResults"]
20:         np_arr = (ws_results['A4'].expand('table')).value
21:         dd = np.array(np_arr).astype(np.float64)
22:         dict_inpts = {}
23:         for keyv,pvalu in zip(dictpos.keys(),new_v.astype(np.float64)):
24:             dict_inpts[keyv] = pvalu
25:         dict_inpts["Type"] = type_v
26:         return dict_inpts,pd.DataFrame(dd)
```

This function outputs two entities: a dictionary containing the parameters inserted to run the simulation and a 4000x10 pandas dataframe with simulation results (which are located inside the "Txl SimResults" tab of the xlsm file). The columns are the ones presented in Table 3.

## 7.2 Generate and save files

To generate the LHS inputs for the NorSandTXL spreadsheet, considering $n\_samples$ soil types and $n\_samples\_2$ initial test conditions, the following code was considered:

```
1:  def gen_NorSand_par_2(dict_ranges_material,dict_ranges_test,n_samples,n_samples_2):
2:      lhs = Lhs(lhs_type="centered", criterion='maximin')
```

```
3:      lhsinner = Lhs(criterion="ratio")
4:      space_material = Space([(0, 1.) for x in range(len(dict_ranges_material))])
5:      space_test = Space([(0, 1.) for x in range(len(dict_ranges_test))])
6:      x_mat = lhs.generate(space_material.dimensions, n_samples,random_state=11)
7:      data_inp_mat = (np.array(x_mat).T)
8:      data_expand_mat = []
9:      for ind_vals in range(len(dict_ranges_material)):
10:        vlow,vup = list(dict_ranges_material.values())[ind_vals]
11:        data_pts = data_inp_mat[ind_vals]
12:        data_expand_mat.append((vup−vlow)∗data_pts + vlow)
13:      data_expand_mat = np.round(np.array(data_expand_mat),4)
14:      data_expand_tst_corretos=[]
15:      for pbb,yv in enumerate(data_expand_mat.T):
16:        x_tst = lhsinner.generate(space_test.dimensions, n_samples_2,random_state=int(11+2∗pbb))
17:        data_inp_tst = (np.array(x_tst).T)
18:        data_expand_tst = []
19:        for ind_vals in range(len(dict_ranges_test)):
20:          if ind_vals==0:
21:            data_expand_tst.append(data_inp_tst[ind_vals])
22:          else:
23:            vlow,vup = list(dict_ranges_test.values())[ind_vals]
24:            data_pts = data_inp_tst[ind_vals]
25:            data_expand_tst.append((vup−vlow)∗data_pts + vlow)
26:        data_expand_tst = np.array(data_expand_tst)
27:        data_expand_tst_prov = data_expand_tst.copy()
28:        data_expand_tst_prov[0] = np.array([(np.clip(yv[2]/(yv[4]∗(1+yv[3])),0, yv[2]/(5∗yv[4]∗(1+yv[3])))+0.2)∗lhsv−0.2 for lhsv in
            data_expand_tst_prov[0]])
29:        data_expand_tst_corretos.append(data_expand_tst_prov)
30:      data_expand_tst_corretos = np.round(np.array(data_expand_tst_corretos),4)
31:      final_comp=[]
32:      for mat_vals,tst_vals in zip(data_expand_mat.T,data_expand_tst_corretos):
33:        for ti_vals in tst_vals.T:
34:          final_comp.append(np.concatenate((mat_vals,ti_vals),axis=0))
35:      return final_comp
```

The quasi-Monte Carlos sampling schemes (Sobol and Halton) can be used to generate the input samples by means of the $gen\_NorSand\_par\_LD$ function, written as:

```
1:  def gen_NorSand_par_LD(dict_ranges_material,dict_ranges_test,n_samples,n_samples_2):
```

```
2:      sampler = qmc.Sobol(d=len(dict_ranges_material), scramble=True,seed=11)

3:      x_mat = sampler.random_base2(m=int(np.log2(n_samples)))

4:      data_inp_mat = x_mat.T

5:      data_expand_mat = []

6:      for ind_vals in range(len(dict_ranges_material)):

7:          vlow,vup = list(dict_ranges_material.values())[ind_vals]

8:          data_pts = data_inp_mat[ind_vals]

9:          data_expand_mat.append((vup−vlow)∗data_pts + vlow)

10:     data_expand_mat = np.round(np.array(data_expand_mat),4)

11:     data_expand_tst_corretos=[]

12:     for pbb,yv in enumerate(data_expand_mat.T):

13:         samplerinner = qmc.Halton(d=len(dict_ranges_test),scramble=True,seed=int(11+2∗pbb))

14:         x_tst = samplerinner.random(n=n_samples_2)

15:         data_inp_tst = x_tst.T

16:         data_expand_tst = []

17:         for ind_vals in range(len(dict_ranges_test)):

18:             if ind_vals==0:

19:                 data_expand_tst.append(data_inp_tst[ind_vals])

20:             else:

21:                 vlow,vup = list(dict_ranges_test.values())[ind_vals]

22:                 data_pts = data_inp_tst[ind_vals]

23:                 data_expand_tst.append((vup−vlow)∗data_pts + vlow)

24:         data_expand_tst = np.array(data_expand_tst)

25:         data_expand_tst_prov = data_expand_tst.copy()

26:         data_expand_tst_prov[0] = np.array([(np.clip(yv[2]/(yv[4]∗(1+yv[3])),0,yv[2]/(5∗yv[4]∗(1+yv[3])))+0.2)∗lhsv−0.2 for lhsv in
             data_expand_tst_prov[0]])

27:         data_expand_tst_corretos.append(data_expand_tst_prov)

28:     data_expand_tst_corretos = np.round(np.array(data_expand_tst_corretos),4)

29:     final_comp=[]

30:     for mat_vals,tst_vals in zip(data_expand_mat.T,data_expand_tst_corretos):

31:         for ti_vals in tst_vals.T:

32:             final_comp.append(np.concatenate((mat_vals,ti_vals),axis=0))

33:     return final_comp
```

On the other hand, to run the NorSandTXL Excel spreadsheet located in $path\_xlsm$ for all the input parameters previously obtained as $final\_comp = gen\_NorSand\_par\_2(\ dict\_ranges\_material,\ dict\_ranges\_test, n\_samples, n\_samples\_2)$ (or $final\_comp = gen\_NorSand\_par\_LD(\ dict\_ranges\_material,\ dict\_ranges\_test, n\_samples, n\_samples\_2)$ for the quasi-Monte Carlo sampling of inputs), the following function can be run:

```python
 1: def run_NorSand_simus_P(final_comp,dictpos,n_samples_2,path_xlsm):
 2:     letras = list(string.ascii_uppercase)
 3:     wb = xw.Book(path_xlsm)
 4:     app = wb.app
 5:     macro_vba = app.macro("'NorTxl.xlsm'!RunSim")
 6:     macro_vba_type = app.macro("'NorTxl.xlsm'!ChangeSimMode")
 7:     ws = wb.sheets["Params_&_Plots"]
 8:     results_comp = []
 9:     for idini,new_v in enumerate(final_comp):
10:         matv = int(math.floor(idini/n_samples_2))
11:         for nv,ps in zip(new_v,dictpos.values()):
12:             pl,pc = ps
13:             pfinal = letras[pc−1]+str(pl)
14:             ws[pfinal].value = nv
15:         for type_v in ["Drained","Undrained"]:
16:             if ws["D34"].value == type_v:
17:                 pass
18:             else:
19:                 macro_vba_type()
20:             macro_vba()
21:             ws_results = wb.sheets["Txl_SimResults"]
22:             np_arr = (ws_results['A4'].expand('table')).value
23:             path_xlsm_init = ("\\").join(path_xlsm.split("\\")[:−1])
24:             new_h5_file = path_xlsm_init+'\\Simus\\'+type_v+"\\Par_"+str(matv)+"_"+str(idini)+".h5"
25:             new_h5_file_spl = new_h5_file.split("\\")
26:             for va in range(−3,0):
27:                 try:
28:                     os.mkdir(os.path.join(∗new_h5_file_spl[:va]))
29:                 except:
30:                     pass
31:             h5f = h5py.File(new_h5_file, 'w')
32:             dd = h5f.create_dataset('NorSandTXL', data=np.array((ws_results['A4'].expand('table')).value).astype(np.float32),compression
       ='gzip')
33:             for keyv,pvalu in zip(dictpos.keys(),new_v.astype(np.float32)):
34:                 dd.attrs[keyv] = pvalu
35:             dd.attrs["Type"] = type_v
36:             h5f.close()
```

The function $run\_NorSand\_simus\_P$ runs the simulation and also saves the results as *.h5* files in the same folder as the Excel spreadsheet. In this case, the new files are saved following the naming convention and folder structure discussed in the paper.

It is worth noticing that for the LHS sampling with 2000 soil types and 40 test conditions, two values of sampled $\psi_0$ needed to be reduced due to instabilities in the VBA code calculations. These were:

- $final\_comp[19572][10] = 0.085$ and

- $final\_comp[10929][10] = 0.082$.

On the other hand, for the quasi-Monte Carlos sampling with 2048 soil types and 42 test conditions, five values of sampled $\psi_0$ needed to be reduced due to the same reasons. These were:

- $final\_comp[56382][10] = 0.0849$,

- $final\_comp[57476][10] = 0.0766$,

- $final\_comp[85371][10] = 0.0955$,

- $final\_comp[34971][10] = 0.08$ and

- $final\_comp[41245][10] = 0.072$.

All the codes previously presented are available as the Jupyter notebook *Sample_and_Run.ipynb* at Ozelim et al. (2023b).

### 7.3   Analyzing errors during learning tasks

As described in the Methods section, we perform a sample size validation. Considering that the codes for such validation are lengthy, they are presented in Ozelim et al. (2023b). The Jupyter notebook *Sample_size_validation.ipynb* is fully commented to illustrate its usage.

## 8   Conclusions

Obtaining massive datasets for modelling the behavior of soils is of great interest, not only because new artificial intelligence algorithms can be built, but also to assess the adequacy of newly proposed physically informed models. In the context of critical state approaches, the NorSand model has shown provide a good balance balance complexity and accuracy. Also, this model is used to assess the liquefaction potential of soils, which is a major cause of high scale disasters lately, such as tailing dams' failures.

In this study, major issues were addressed. Firstly, the paper tackled the challenges associated with the quantity and complexity of synthetic datasets required for nonlinear constitutive modeling of soils. This was achieved by simulating both drained and undrained triaxial tests, resulting in two datasets. The first dataset involved a nested Latin Hypercube Sampling of input

parameters, covering 2000 soil types with 40 initial test configurations for each, yielding a total of 160000 triaxial test results. The second dataset employed a nested quasi-Monte Carlo sampling (Sobol and Halton) of input parameters, encompassing 2048 soil types with 42 initial test configurations for each, resulting in a total of 172032 triaxial test results. Each simulation dataset was represented as a matrix of dimensions $4000 \times 10$. The study demonstrated that the dataset of 2000 soil types and 40 initial test configurations adequately captured the general behavior of the NorSand model.

Secondly, the paper addressed the issue of the availability of open-source implementations of the NorSand constitutive model. This was achieved by presenting an implementation that connects the well-established VBA implementation to the Python environment. The VBA code served as the "processing kernel" for the new Python implementation, leveraging the extensive testing and validation conducted by Jefferies and Been (2015). This integration allows researchers to harness the full capabilities of Python packages in their analyses involving the NorSand model.

A comprehensive database like the one provided is crucial for developing ML and artificial intelligence models of geotechnical materials. We are confident that this database will be widely used by both academic and industry communities. Furthermore, researchers interested in modeling sequential data, such as time series, could use this dataset for benchmarking purposes, as the highly non-linear nature of the constitutive model poses a significant challenge to ML and DL techniques.

## 9   Code and data availability

All data associated with the current submission is available at Ozelim et al. (2023a). Any updates will also be published on Zenodo, and the final DOI cited in the manuscript. The Python code used to generate the NorSandAI dataset is described in the present paper and available at Ozelim et al. (2023b). Besides, the codes used for the learning task considered are also available at Ozelim et al. (2023b)

*Author contributions.*  Conceptualization, L.C.d.S.M.O.; methodology, L.C.d.S.M.O.; software, L.C.d.S.M.O.; validation, L.C.d.S.M.O.; formal analysis, L.C.d.S.M.O., M.D.T.C. and A.L.B.C.; investigation, L.C.d.S.M.O.; writing—original draft preparation, L.C.d.S.M.O.; writing—review and editing, M.D.T.C. and A.L.B.C.; supervision, M.D.T.C.; funding acquisition, L.C.d.S.M.O. All authors have read and agreed to the published version of the manuscript.

*Competing interests.*  The authors declare no competing interests.

*Acknowledgements.*  The authors acknowledge the support of the National Council for Scientific and Technological Development (CNPq Grant 102414/2022-0). This study was also financed in part by the Coordination for the Improvement of Higher Education Personnel (CAPES)—Finance Code 001.

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
