# Peer review of "NorSand4AI: A Comprehensive Triaxial Test Simulation Database for NorSand Constitutive Model Materials"

_EGUsphere, 2023_

## Author Response (AR1)

**Authors' Response to Reviews of**

**NorSand4AI: A Comprehensive Triaxial Test Simulation Database for NorSand Constitutive Model Materials**

Ozelim et al.

*Geoscientific Model Development - GMD,* `egusphere-2023-1690`
* * *
RC: *Reviewers' Comment*,     AR: Authors' Response,     ☐ Manuscript Text

**Dear Prof. Dr. Le Yu, Handling Topic Editor of GMD**

May this letter find you well.

We thank the valuable comments provided by the reviewers. A complete reply to each of the questions raised is hereby presented. The changes suggested by the reviewers were crucial to enhance the quality of the paper. One may notice, for example, that the manuscript total length went from 13 to 25 pages. We hope you find our manuscript suitable for publication and look forward to hearing from you in due course.

**1.   Reviewer #1**

RC:   *Dear authors,*

This work aims to provide a useful synthetic dataset in assessing the liquefaction potential of soils for machine learning and deep learning tasks. I consider that more validation and scrutiny are required for such essential groundwork.

Table 1 shows the sampling range of the Nor-Sand model's input. However, justifications for such ranges are not mentioned in the manuscript. In the conclusion section, the authors noted that the Nor-sand model is used to assess the liquefaction potential of soils. The utility of this work should be highlighted throughout the article (not just in the conclusion session) to identify the target audience better and improve readership. Then, the ranges of the parameters can be justified.

AR:   Dear reviewer,

About Table 1, indeed the original preprint lacked some clarifications on why the ranges adopted were of interest. The ranges adopted come from literature results on the behavior of real granular materials. An initial version of such ranges was first presented by Jefferies and Shuttle (2002) and has been updated ever since. The ranges presented in the paper are based on the latest compilation available, thus Table 1 reflects the information presented in the book by Jefferies and Been (2015). As mentioned, those authors have collected several triaxial tests carried out on a diverse set of granular soils, which eventually led to the creation of Table 1. In the updated version of the manuscript, we will include such information, describing why the values presented are of interest. The following text has been inserted right before Table 1

> The input parameters of the NorSand model are presented in Table 1. The sampling ranges adopted

> come from literature results on the behavior of real granular materials. An initial version of such ranges was first presented by Jefferies and Shuttle (2002) and has been updated ever since. The ranges presented in Table 1 reflect the latest compilation available and reported by Jefferies and Been (2015). This way, practitioners will especially benefit from the datasets generated, since the parameters involved have been chosen as to represent real granular materials. Table 1 also present the meaning of each parameter in the column "Description".

Regarding the application to liquefaction modelling, indeed we had not presented this aspect throughout the article. In the updated version we completely reformulated the introduction to account for that, highlighting how the NorSand is used in that context to make the reader aware of the benefits of our approach. These changes can be seen on Sections 1 and 3, which are a completely rewritten Introduction and a section dedicated to the NorSand model, respectively.

**RC:** *The output of the Nor-Sand model spreadsheet can be technically sound. However, the authors should test the sensitivity of the sampled dataset. For example, why would such sampling be the best dataset to represent the Nor-Sand model? Can one represent the model better with fewer samples, or more samples are required? Without showing a particular use (e.g., surrogate modeling, machine learning) or arguing the representativeness of the dataset, it is difficult to evaluate the value of such a dataset.*

**AR:** We are glad the reviewer pointed that out. At first, we performed a number of empirical simulations to check how big would the dataset be in order to represent the true behavior of the NorSand model. For simplicity, we ended up no including these studies in the manuscript.

On the other hand, after reading the comments, we had to devise a proper methodology (and not present just a series of empirical tests) to demonstrate in a robust and reproducible way that the dataset presented suffices to represent the NorSand model. This way, a completely new methodology has been proposed and applied to assess the quality of the sample size.

As suggested by the reviewer, the best way to show that the sample size is sufficient is to study how a model calibrated (or trained) on such dataset performs. So, we chose the most direct (and actually most important) learning task one could face while working with the dataset generated: back-calculation of the constitutive parameters of the model based solely on the triaxial test results. In short, from the triaxial tests we will learn the values of the parameters which govern the behavior of the material.

Section 4.2 of the updated draft presents all the details of the new framework considered. It now reads:

[revised manuscript text omitted]
 results of applying this new methodological framework to assess the sufficiency of the datasets led to the results now presented in Section 6 of the updated draft, which follow below:

**6. Technical Validation**

Considering that the engine running the triaxial test simulations is the Excel spreadsheet presented in the book by Jefferies and Been (2015) a and that such spreadsheet has been extensively validated by both academia and industry, there is no need to discuss the technical quality of the dataset. On the other hand, it is necessary to show that $In_{2000,40}$ suffices to cover the general behavior of the NorSand models.

By following the methods previously described and plotting the mean absolute percentage error (MAPE) result of the 49 models (each trained and validated with samples of different sizes subsampled from $qIn_{2048,42}$) Figure 2 and 3 were obtained for drained and undrained conditions, respectively. The 4 algorithms considered were Ridge, KNeighbors, Ridge-K (with nonlinear kernel on inputs) and Ridge-KT (with nonlinear kernel on inputs and also QuantileTransformer on the outputs). It is clear in the figures that, for contours of 0.5% gains in MAPE, the sample size of 2000x40 is actually more than enough for the learning task considered. This can be stated by noticing that the contours with lower error encompass samples with an exponential range of sizes (the x-axis is in log scale). This indicates a really small gradient on the error in the $nxm$ space, implying a good sample size. This happens for all 4 algorithms, indicating that not only linear and neighbors-based regressors have reached their maximum ability to learn, but also the nonlinear variants considered. It can be seen that the two nonlinear transformations applied (to inputs and to both inputs and outputs) present a similar behavior, although with considerably smaller MAPEs.

Due to the space-filling qualities of both $In_{2000,40}$ and $qIn_{2048,42}$, $qIn_{2048,42}$ can also be considered a sufficient dataset to represent the NorSand model.

RC: *Since there are too many possible ways to improve the manuscript, I leave the authors to decide which aspects they would like to work on. I do not recommend the manuscript for publication at this stage.*

AR: We sincerely thank the reviewer for pointing out such interesting and important issues. We believe the updated version of the manuscript covers all the issues raised, and hope the paper is now suitable for publication. The new dataset as well as the codes used to perform the analyses are all available in Zenodo.

**2.  Reviewer #2**

RC: *General comments This paper tried to establish a comprehensive triaxial test simulation database for soil science. It is attractive to develop this kind of model, but I have some questions about this approach. Especially, what is the advantage improved from the previous approach needs to be clearly introduced and explained in this study. In addition, for better presentation quality, I strongly suggest reorganizing the manuscript because the current manuscript contains so many paragraphs. Please address the following questions.*

AR: Dear reviewer, We recognize the structure of the paper needed enhancement. In the updated version we incorporated the suggestions presented. About the paper, overall, there are two main advantages of using our results and datasets. The first one is that there are no known implementations of the NorSand model in Python. So, we built a bridge which connects a well-known VBA implementation to the Python environment. This allows other researchers to consider our code as a step in their Pipelines, allowing them to use the full power of Python packages (such as sklearn, TensorFlow, Pytorch etc) during their analyses. This has been included in the manuscript as:

[Figure]

Figure 2: Mean absolute percentage error for all the 14 parameters after being back-calculated solely from drained triaxial test results.

Also, only recently the NorSand method has been implemented in commercial Finite Element softwares (Rocscience, 2022; Itasca Consulting Group, 2023; Bentley, 2022). Besides, regarding open-source distributions, only the Visual Basic (VBA) implementation presented by Jefferies and Been (2015) is available. Thus, another open-source implementation easily integrated into ML and DL modelling pipelines is desirable. [...] Thus, the current paper aims to address three main issues: the quantity and complexity of synthetic datasets for nonlinear constitutive modeling of soils and the availability of open-source implementations of the NorSand constitutive model. [...] Then, the third aspect is considered by presenting an implementation which connects the well-known VBA implementation to the Python environment. We will use the VBA code as the "processing kernel" of our Python implementation, taking advantage of the years of tests and validation of the algorithm provided by Jefferies and Been (2015). This new Python code allows other researchers to use the full power of Python packages during their analyses involving NorSand.

The second advantage is that, each evaluation of the NorSand Model in the VBA code takes some time and effort to be completed. So, by providing massive simulation results, we save a considerable number of hours (even days) from other researchers which need such datasets.

[Figure]

Figure 3: Mean absolute percentage error for all the 14 parameters after being back-calculated solely from undrained triaxial test results.

**RC:** *Specific comments*

Many paragraphs throughout the manuscript: Repeatedly, there are lots of paragraphs, and some paragraphs seem to be merged. Please reorganize for better readability.

AR: We completely restructured the paper to account for such suggestion. The reviewer may check that all the section had insertions/changes.

**RC:** *L26-34: These two paragraphs started with 'Montans et al. (2019) emphasize...' and ended with '(Montans et al., 2019)'. It is unclear what is authors' statements and what is referred statements. These two paragraphs may be merged.*

AR: Indeed, the paragraphs could be merged and the reference could be better placed. We implemented that in the new version of the manuscript.

**RC:** *L57: From here, the introduction is suddenly changed to soil science. To fill the gap in the general introduction, some background information for soil science will be needed.*

AR: Indeed, the paper lacked a more comprehensive review on soil sciences. We added such information in the

updated version. Besides, a whole new section (Section 3 in the updated draft) has been created to discuss the NorSand model. We believe the new introduction is more complete and thank the reviewer for pointing out its previous deficiencies.

**RC:** *L57: In addition to the above comment, it is hard to follow these previous studies. One idea is to prepare a brief summary for a clear introduction. Please reconsider.*

**AR:** This is a nice suggestion. We restructured the introduction to account for summaries on previous studies, focusing on soil sciences and liquefaction assessment.

**RC:** *Table 1: A brief description of these parameters will be helpful for readers. The abbreviation of "OCR" should be explicitly defined within this manuscript. This might be covered within the previous studies, but why the sampling range can be set as listed in Table 1? Even though NorSandTXL has been already described in previous studies, a kinder introduction for Table 1 is required because this manuscript itself should be standalone.*

**AR:** About Table 1, indeed the original preprint lacked some clarifications on why the ranges adopted were of interest and also what each parameter means. For example, OCR stands for over consolidation ratio. A novel section (Section 3) has been added to the manuscript to better situate the reader with respect to the NorSand model. Besides, a new column ("Description") has been added to Table 1, clearly indicating the meaning of each parameter. The ranges adopted come from literature results on the behavior of real granular materials. An initial version of such ranges was first presented by Jefferies and Shuttle (2002) and has been updated ever since. The ranges presented in the paper are based on the latest compilation available, thus Table 1 reflects the information presented in the book by Jefferies and Been (2015). As mentioned, those authors have collected several triaxial tests carried out on a diverse set of granular soils, which eventually led to the creation of Table 1. In the updated version of the manuscript, we included include such information, describing why the values presented are of interest as well as what each parameter controls in the soil's behavior. This specific insertion reads as:

> The input parameters of the NorSand model are presented in Table 1. The sampling ranges adopted come from literature results on the behavior of real granular materials. An initial version of such ranges was first presented by Jefferies and Shuttle (2002) and has been updated ever since. The ranges presented in Table 1 reflect the latest compilation available and reported by Jefferies and Been (2015). This way, practitioners will especially benefit from the datasets generated, since the parameters involved have been chosen as to represent real granular materials. Table 1 also present the meaning of each parameter in the column "Description".

**RC:** *L156-159 and L170-171: It is still unclear why this Python coding is needed and excel spreadsheet is not acceptable. This point seems to be argued in L102-107, but for practical use, how about calculating time by spreadsheet and Python, or how about the operational advantages?*

**AR:** About the Python coding, it is not that the excel spreadsheet is not acceptable. A first thing to notice is that there are no known implementations of the NorSand model in Python. So, we built a bridge which connects a well-known VBA implementation to the Python environment. In the end, we still rely on the VBA code as the "processing kernel" of our Python implementation. This new Python code allows, on the other hand, other researchers to use the full power of Python packages (such as sklearn, TensorFlow, Pytorch etc) during their analyses involving NorSand. The second advantage is that, each evaluation of the NorSand Model in the VBA code takes some time and effort to be completed (setting parameters, choosing simulation type, running the VBA macros and collecting results). So, by providing massive simulation results, we save a considerable number of hours (even days) from other researchers which need such datasets. The text added to account for

this specific issue was presented in the response to the first "General comment".

**RC:** *L160 (Section 5): I was impressed that this section seems to be moved to the Appendix part.*

AR: We chose to move the coding part to the Appendix because we wanted to focus on the datasets in the main "body" of the manuscript. But we moved the codes back from the appendix and insert them in Section 7 (previously Section 5), as suggested. Besides, we included additional codes in the manuscript and created a github repo to make their sharing easier. We incorporated a new simplified code which simply outputs the simulation values instead of directly saving them to a .h5 file. This will make the incorporation of the code into existing Pipelines easier.

**RC:** *Technical comments L51: Use "DNN". L89: No need to repeat "(Jefferies, 1993)" here. L105: Please insert after Table 1, and there maybe no need to change the paragraph here. Tables 1, 2, and 3: The caption should be placed at the top of the table.*

AR: We incorporated all the issues above in the final version of the manuscript. We sincerely thank the reviewer for such careful analysis on the paper, specially the code parts.

Sincerely,

Luan Carlos de Sena Monteiro Ozelim, D.Sc.
Corresponding author on behalf of all authors

---

## Referee Report (RR1)

**Review of:**

**NorSand4AI: A Comprehensive Triaxial Test Simulation Database for NorSand Constitutive Model Materials**

…submitted for publication in Geoscientific Model Development (an open-access journal of the European Geosciences Union).

**General Comments**

The ms offers an approach to using AI for assessing soil behaviour, with a focus on static liquefaction. This is novel. But, the nature of this novel contribution is unclear in both the Abstract and the Introduction – the key ideas do not occur until L62-68, with further relevant comments in L128-133. Further, the Abstract is too long with more detail than appropriate while not highlighting the main ideas and contribution.

More generally, the ms focuses on details of the process rather than results. For example, there is no figure comparing the AI output with the NS model used for example soil properties or, better, several examples). This leaves an interested reader uncertain on how well the AI offered works, with Figures 2 and 3 not helping if you are not already familiar with details of AI.

A related concern is the range of soil properties over which the AI has been trained. Table 1 provides the ranges used, and most seem reasonable (or at least in accord with what might be expected for 'sands'). There are issues with these ranges as it appears each soil property has been treated as independent whereas $H_0$ is commonly inversely correlated with $\lambda$ and also directly correlated with G – if the soil is stiff elastically, then it will be stiff plastically. Thus, there appear to be property combinations that are not physically representative. Further, in the case of OCR the lower limit is unity, not 0.5: all critical state models do not allow under-consolidation.

These concerns lead me to the conclusion that the paper needs re-writing to properly present the ideas, likely moving some detail to an appendix, with an expanded presentation to illustrate the performance of AI as it might be utilized by a practicing engineer. Additional comments follow to assist this reworking on the ms.

**Detailed Comments**

L63. All critical state models have particular idealizations, the most obvious of which is that all of them (at least to date) are based on 'remoulded' (or disturbed) soil properties. You might argue that 'structure' might appear as apparent OCR (a common view with clays, see Bjerrum's Rankine Lecture) but then where does that leave $\psi$ ? Should there be a cohesion component to the strength ? Could M vary with strain ? These points are not a criticism of your approach as such, but rather aspects that need identifying to the reader. Indeed, and this is something you might discuss in a revised ms, it seems to me your AI could be quickly

used to assess a new set of data to evaluate if indeed 'structure' might be something that needed considering.   If I understand correctly, you allude to this in L128-133.

L83.  An open-source Python implementation is an excellent idea/contribution, but it really is a different subject than AI.  Present the Python work as a second paper, not necessarily to this journal.

L161.  NS is closer to original cam clay (OCC; Schofield & Wroth) than MCC, with NS and OCC yield surfaces having the same shape and the same flowrule.

L177/Table 1.  The sampling ranges used really should be presented as a distinct new section, as it is a new topic. Indeed, you might even move it forward to the discussion of training NN as these properties are not intrinsic to NS (with the exception of H, which you could slave to $\lambda$, say using $H_0 = 2/\lambda$).  I also wonder if there should not be a figure, say plotting $\chi$ vs $\lambda$ with the points using a different symbol for M – not comprehensive, but it would illustrate the space occupied by your realized 'training' cases; such a figure might usefully follow L195-208.

L231.  I do not understand what comprises your dataset developed using NorSandTXL.  Is it a set of strains and stresses, and if so at what strain increments ?  Are you capturing what amounts to a numerical triaxial tests with 100 steps ?  300 steps ?  I am guessing that a 'dataset' amounts to an array [n,4];  this needs clarifying.

L288.  This needs a figure illustrating the q-e1 with 40 vs 4000 points.  This is a quite extreme compression to my eyes, as 40 points always seems too few when recording a lab test.  And please indicate whether you run out to 10%, 15% or what strain, keeping in mind the critical state is a large-strain condition.  I am also surprised that you regress on e – I would have use the dilatancy; and, it may be appropriate to treat drained and undrained tests differently.

Section 6, Validation.  I found this section unconvincing.  The aim is to recover best-estimate soil properties from a [40,3] reduced dataset.  There are 10 properties and 4 state measures (Table 3).  Actually, thinking as I type you could remove $\nu$ as it is rarely measured and commonly just assumed as a "not unreasonable" value.  What is needed are plots showing 'truth' (= known property of training dataset) on x-axis vs 'prediction' (= recovered using AI) on y axis; I would focus on just a few properties/state to keep the ms to a reasonable size, say:  $\lambda$, $\chi$, G, $\psi$.  Such plots will allow a reader to quickly assimilate how well the procedures presented in the paper work.

---

## Author Response (AR2)

**Authors' Response to Reviews of**

**NorSand4AI: A Comprehensive Triaxial Test Simulation Database for NorSand Constitutive Model Materials**

Ozelim et al.

*Geoscientific Model Development - GMD,* `egusphere-2023-1690`
* * *
RC: *Reviewers' Comment*,     AR: Authors' Response,     ☐ Manuscript Text

**Dear Prof. Dr. Le Yu, Handling Topic Editor of GMD**

May this letter find you well.

Once again, we thank the valuable comments provided by the reviewers. A complete reply to each of the questions raised is hereby presented. We were glad to receive the comments by Referee #3 (Report #2), as it was the first time he reviewed the manuscript and his suggestions were quite interesting. Overall, all the changes suggested by both reviewers were crucial to enhance the quality of the paper. We hope you find our manuscript suitable for publication and look forward to hearing from you in due course.

**1. Reviewer #2 - Report #1**

RC: *I appreciate the authors' many works to revise the original manuscript fully. All of my comments have been well solved, and the quality of the manuscript has been improved. Now I would like to recommend the publication. However, regarding the newly added parts, please consider the following minor comments before the final publication.*

AR: We are glad the changes undertaken were sufficient to address the issues raised in the first round of reviews. We sincerely thank the reviewer for writing both reports on the paper. The minor issues indicated were all addressed, as shall be seen below.

RC: *Abstract, L3: As a short introduction of the model name, how about expressing "parametric models known as constitutive models (e.g., the Modified Cam Clay and the NorSand) are used to ..." here?*

AR: We inserted the text as suggested by the reviewer.

RC: *L133: Section 2.1.1 could be Section 2.2.*

AR: We changed the numbering accordingly.

RC: *L160, L163, and L171: It is better to replace "Modified Cam Clay" with "MCC" because MCC has been already defined in L138.*

AR: We agree and changed the text as requested.

RC: *L575: Please double-check this sentence "balance balance".*

AR: Indeed the word "balance" was repeated for no reason. We corrected that.

**RC:** *Figures 2 and 3: We can understand the implications for the learning task from this figure, but it could be much better to unify all color-scale for better comparison between four methods and drained/undrained conditions. What is a possible reason for a slightly increasing error by the Ridge-K and Ridge-KT model in the right-bottom (large constitutive parameters samples with lower test parameter samples)? In addition, why does the undrained triaxial test generally lead to better performance (i.e., small error) compared to the drained triaxial test?*

**AR:** This is a great suggestion. We unified the color-scale of all the plots and added the numerical labels to each contour. We think the new version of the plots is much better to read and perform comparisons.

About the slightly increasing error by the Ridge-K and Ridge-KT model in the right-bottom (large constitutive parameters samples with lower test parameter samples), we understand that visually this would seem to be true, but in fact it is an artifice of the log-scale on the x axis (which compressed the values on that corner). If natural scale was considered, you would see that the opposite occurs (large constitutive parameters samples with lower test parameter samples give better results while compared to small constitutive parameters samples with large test parameter samples). Such behavior occurs because out of the 14 parameters, 10 correspond to constitutive parameters, so less training samples impair their learning task. This behavior can be seen on Figures 17 and 18, newly added to avoid this visual confusion. There, it is possible to see that the errors of the 10 constitutive parameters are much more sensitive to less training samples than the opposite situation with test parameters. Except for $OCR$, all the other heavily impaired parameters are constitutive ones. A whole new subsection has been added to explain this (section 6.1.2), which reads:

> By analyzing Figures 5 and 6, apparently the overall MAPE slightly increases in the right-bottom corner (large constitutive parameters samples with lower test parameter samples). This is a visual artifice caused by the application of the log-scale to the horizontal axis, which ends up compressing the values on that corner. If the natural scale was considered, one would see that the opposite occurs: large constitutive parameters samples with lower test parameter samples give better results while compared to small constitutive parameters samples with large test parameter samples. Such behavior can be explained by noticing that out of the 14 parameters, 10 correspond to constitutive parameters, so less training samples impair their learning task.
>
> A MAPE comparison is presented in Figures 17 and 18 for both drained and undrained tests with different training sample's diversities (we compare the best performing models obtained by Ridge-KT algorithm, which use the $2048 \times 42$ dataset, to two other case: $32 \times 42$ and $2048 \times 6$ training samples. It is possible to see that the errors of the 10 constitutive parameters exhibit a greater sensitivity to less training samples than the opposite situation with test parameters. Except for $OCR$, all the other heavily impaired parameters are constitutive ones.

About the reason why undrained tests generally lead to better performance compared to drained tests, we believe this happens because during undrained tests the void ratio is constant. Thus, for the learning task provided, the algorithm does not need to perform any nonlinear operations on one third of the input dataset (which consists of $e$, $p$ and $q$ for 40 values of $\epsilon_1$). So, with the same number of training samples and analytical structure of the learning algorithm, it is understandable that less nonlinearities in the inputs would result in a better performance (smaller errors) of the predicted outputs. To account for such explanation, we inserted the following text right after Figures 5 and 6:

> The analysis of Figures 5 and 6 indicate that for the learning task hereby considered, undrained tests generally presented a better performance while compared to drained tests. A possible cause for such

behavior is that during undrained tests the void ratio is kept constant. Thus, for the learning task considered, the algorithm does not need to perform any nonlinear operations on one third of the input dataset (which consists of $e$, $p$ and $q$ for 40 values of $\epsilon_1$). So, with the same number of training samples and analytical structure of the learning algorithm, it is expected that less nonlinearities in the inputs would result in a better performance (smaller errors) of the predicted outputs.

RC: *Throughout the manuscript: Some parts use "x" instead of the $\times$, especially in the expression of the metrics. Please carefully check.*

AR: We carefully reviewed the whole manuscript and changed all the "x" to $\times$ .

Once again, we are grateful for all the suggestions indicated.

**2. Reviewer #3 - Report #2**

RC: *General Comments The ms offers an approach to using AI for assessing soil behaviour, with a focus on static liquefaction. This is novel. But, the nature of this novel contribution is unclear in both the Abstract and the Introduction – the key ideas do not occur until L62-68, with further relevant comments in L128-133. Further, the Abstract is too long with more detail than appropriate while not highlighting the main ideas and contribution.*

AR: Dear reviewer, we are glad to receive new comments about the manuscript. We will address them to the best of our capabilities, trying to make a good balance between all the suggestions presented by all the three reviewers.

About the abstract, indeed it was too long lacked a more precise presentation of the research ideas and results. This way, the abstract has been partically rewritten and now reads:

In soil sciences, parametric models known as constitutive models (e.g., the Modified Cam Clay and the NorSand) are used to represent the behavior of natural and artificial materials. In contexts where liquefaction may occur, the NorSand constitutive model has been extensively applied by both industry and academia due to its relatively simple critical state formulation and low number of input parameters. Despite its suitability as a good modelling framework to assess static liquefaction, the NorSand model still is based upon premises which may not perfectly represent the behavior of all soil types. In this context, the creation of data-driven and physically-informed metamodels emerges. Literature suggests that data-driven models should initially be developed using synthetic datasets to establish a general framework, which can later be applied to experimental datasets to enhance the model's robustness and aid in discovering potential mechanisms of soil behavior. Therefore, creating large and reliable synthetic datasets is a crucial step in constructing data-driven constitutive models. In this context, the NorSand model comes handy: by using NorSand simulations as the training dataset, data-driven constitutive metamodels can then be fine-tuned using real test results. The models created that way will combine the power of NorSand with the flexibility provided by data-driven approaches, enhancing the modelling capabilities for liquefaction. Therefore, for a material following the NorSand model, the present paper presents a first-of-its-kind database that addresses the size and complexity issues of creating synthetic datasets for nonlinear constitutive modeling of soils by simulating both drained and undrained triaxial tests. Two datasets are provided: the first one considers a nested Latin Hypercube Sampling of input parameters encompassing 2000 soil types, each subjected to 40 initial

test configurations, resulting in a total of 160000 triaxial test results. The second one considers nested quasi-Monte Carlos sampling techniques (Sobol and Halton) of input parameters encompassing 2048 soil types, each subjected to 42 initial test configurations, resulting in a total of 172032 triaxial test results. By using the quasi-Monte Carlo dataset and 49 of its subsamples, it is shown that the dataset of 2000 soil types and 40 initial test configurations is sufficient to represent the general behavior of the NorSand model. In this process, four Machine Learning algorithms (Ridge Regressor, KNeighbors Regressor and two variants of the Ridge Regressor which incorporate nonlinear Nystroem kernels mappings of the input and output values) were trained to predict the constitutive and test parameters based solely on the triaxial test results. These algorithms achieved 13.91 % and 16.18 % mean absolute percentage errors among all the fourteen predicted parameters for undrained and drained triaxial tests inputs, respectively. As a secondary outcome, this work introduces a Python script that links the established VBA implementation of NorSand to the Python environment. This enables researchers to leverage the comprehensive capabilities of Python packages in their analyses related to this constitutive model.

Regarding the Introduction, we recognize the same issues. Therefore, we rewrote its beginning and ending to provide a more to-the-point presentation of our research objectives and general methods.

**RC:** *More generally, the ms focuses on details of the process rather than results. For example, there is no figure comparing the AI output with the NS model used for example soil properties or, better, several examples). This leaves an interested reader uncertain on how well the AI offered works, with Figures 2 and 3 not helping if you are not already familiar with details of AI.*

AR: We agree that the results of the machine learning algorithms trained needed to be better presented and discussed. Therefore, we included a new subsection (6.1) to provide an in-depth discussion about our findings. Regarding the comparison of the outputs of the algorithms to the real outputs (which were used as parameters of the NorSand model), the newly added Figure 7 presents the individual MAPE for each parameter estimated. Also, ten new figures have been inserted in this particular subsection to illustrate the discussions considered. We believe the results are way clearer now.

**RC:** *A related concern is the range of soil properties over which the AI has been trained. Table 1 provides the ranges used, and most seem reasonable (or at least in accord with what might be expected for 'sands'). There are issues with these ranges as it appears each soil property has been treated as independent whereas $H_0$ is commonly inversely correlated with $\lambda$ and also directly correlated with $G$ - if the soil is stiff elastically, then it will be stiff plastically. Thus, there appear to be property combinations that are not physically representative. Further, in the case of OCR the lower limit is unity, not 0.5: all critical state models do not allow under-consolidation.*

AR: We understand the issues raised about the independent sampling of input parameters. This was done on purpose, to "sample" the behavior of the NorSand model along all possible regions of input parameters' space. This is common in Machine/Deep Learning and is carried out to provide a greater knowledge of the transfer function which takes the parameters as inputs and then outputs the triaxial tests. This is done to ensure the learning process is not biased, in the sense we are not teaching the algorithm just the particular behavior in the regions of interest. For example, if our transfer function was quadratic, depending on the specific subspace we train the algorithms, we could learn a linear relation. For sure the linear relation would work locally, but the general analytical behavior of the solution would not be learnt. Sometimes, learning outside these regions can have a positive impact on the learning process altogether. For particular applications, where this correlation of input parameters is more determinant, different loss weights could be added for inside and outside points. This is, on the other hand, a choice than can be made. In future works, since we will build constitutive models

for specific purposes, this correlation structure will for sure be considered. For the learning task considered in the present paper, and for a more general approach, we understand this is not mandatory.

To make this choice clear, the following paragraphs has been added to the manuscript:

> One may notice that besides $\psi_0$, which is restricted by a fraction of $\psi_{\max}$, an independent sampling of input parameters was conducted. This was considered to explore the behavior of the NorSand model across all conceivable regions of the input parameter space. The objective was to enhance understanding of the analytical characteristics of the transfer function, which accepts these parameters as inputs and produces triaxial test results as outputs. This strategy ensures that the learning process remains unbiased, thereby preventing the algorithm from solely learning the transfer function within a specific area of interest. Broadening the scope of learning task beyond such confines can positively influence the overall learning process. For specific applications where the correlation among input parameters holds greater significance, adjusting loss weights for points within and outside the region of interest could be beneficial. This adjustment represents a choice that can be made. In future works, especially in the development of constitutive models tailored for specific purposes, it is advisable to consider this correlation structure.

RC:     *These concerns lead me to the conclusion that the paper needs re-writing to properly present the ideas, likely moving some detail to an appendix, with an expanded presentation to illustrate the performance of AI as it might be utilized by a practicing engineer. Additional comments follow to assist this reworking on the ms.*

AR:     We thank the reviewer for his suggestions and, as indicated, we rearranged the paper and included section 6.1 to provide an in-depth discussion of the AI results obtained.

RC:     *Detailed Comments L63. All critical state models have particular idealizations, the most obvious of which is that all of them (at least to date) are based on 'remoulded' (or disturbed) soil properties. You might argue that 'structure' might appear as apparent OCR (a common view with clays, see Bjerrum's Rankine Lecture) but then where does that leave $\psi$ ? Should there be a cohesion component to the strength ? Could $M$ vary with strain ? These points are not a criticism of your approach as such, but rather aspects that need identifying to the reader. Indeed, and this is something you might discuss in a revised ms, it seems to me your AI could be quickly used to assess a new set of data to evaluate if indeed 'structure' might be something that needed considering. If I understand correctly, you allude to this in L128-133.*

AR:     We agree such discussions are really important and quite needed. We believe, on the other hand, they are more suitable to be included in future works, when experimental data are used to fine-tune the models developed. In the current manuscript, we put out main focus on the synthetic databases and codes. On the other hand, we included the following text into the Conclusions section, bringing such challenges into consideration and suggesting future studies.

> A comprehensive database like the one provided is crucial for developing ML and artificial intelligence models of geotechnical materials. In particular, all geotechnical critical state models involve specific simplifications, with the most apparent being their reliance on 'remoulded' or disturbed soil properties. Understanding the consequences of such structural alterations, especially in terms of their impact on the apparent $OCR$, poses notable challenges. The effect on the stress ratio ($\psi$) remains unclear. Through the utilization of physics-informed machine learning and artificial intelligence algorithms, these uncertainties can be thoroughly investigated, uncovering patterns and hidden features within experimental data. We are confident that the results of the present paper are useful assets in this quest,

being useful for both academic and industrial communities. Furthermore, researchers interested in modeling sequential data, such as time series, could use this dataset for benchmarking purposes, as the highly non-linear nature of the constitutive model poses a significant challenge to ML and DL techniques.

**RC:** *L83. An open-source Python implementation is an excellent idea/contribution, but it really is a different subject than AI. Present the Python work as a second paper, not necessarily to this journal.*

**AR:** We are glad the reviewer found our contribution interesting. We tried to re-shape the introduction to make it clearer that our main contributions are the databases and the code. We also mentioned the AI learning task we considered as a by-product, but we believe we should keep the Python codes in the current manuscript. Future works will dive deeper into the AI models and how to enchance them.

**RC:** *L161. NS is closer to original cam clay (OCC; Schofield & Wroth) than MCC, with NS and OCC yield surfaces having the same shape and the same flowrule.*

**AR:** We added this info into the manuscript.

**RC:** *L177/Table 1. The sampling ranges used really should be presented as a distinct new section, as it is a new topic. Indeed, you might even move it forward to the discussion of training NN as these properties are not intrinsic to NS (with the exception of $H$, which you could slave to $\lambda$, say using $H_0 = 2/\lambda$). I also wonder if there should not be a figure, say plotting $\chi$ vs $\lambda$ with the points using a different symbol for $M$ - not comprehensive, but it would illustrate the space occupied by your realized 'training' cases; such a figure might usefully follow L195-208.*

**AR:** We agree that the sampling ranges are not intrinsic to the NS model. Thus, we moved the text where they were described to the next section (Data Generation inside the Methods Section). We also updated the ending of Section 3 to account for that, as suggested.

We included Figure 1 with some plots to show the sampling process, as suggested.

**RC:** *L231. I do not understand what comprises your dataset developed using NorSandTXL. Is it a set of strains and stresses, and if so at what strain increments ? Are you capturing what amounts to a numerical triaxial tests with 100 steps ? 300 steps ? I am guessing that a 'dataset' amounts to an array [n,4]; this needs clarifying.*

**AR:** We agree the presentation of the manuscript was a bit strange. We moved some of the content of Section 5 (Data Records) to subsection 4.1 (Data Generation). We believe it is now clearer which of the outputs of the NorSandTXL spreadsheet are considered (Please see Table 2). We consider the 4000 increments provided by the NorSandTXL.

**RC:** *L288. This needs a figure illustrating the q-e1 with 40 vs 4000 points. This is a quite extreme compression to my eyes, as 40 points always seems too few when recording a lab test. And please indicate whether you run out to 10%, 15% or what strain, keeping in mind the critical state is a large-strain condition. I am also surprised that you regress on e – I would have use the dilatancy; and, it may be appropriate to treat drained and undrained tests differently.*

**AR:** We included Figures 3 and 4 to show the downsampling process. Both the trimming (make all results have the same overall range) as well as downsampling with 40 points logarithmically separated are illustrated.

We chose to regress on $e$ as it is common to obtain triaxial test results using this parameter. On the other hand, other studies may be carried out and, indeed, could provide better results. We added the following remark

when explaining Figure 2:

> The other 7 columns are manipulations of these three ($D_p$ or $\eta$, for example) and could be used as alternative regression variables, but such selection is not the focus of the present paper.

**RC:** *Section 6, Validation. I found this section unconvincing. The aim is to recover best-estimate soil properties from a [40,3] reduced dataset. There are 10 properties and 4 state measures (Table 3). Actually, thinking as I type you could remove $\nu$ as it is rarely measured and commonly just assumed as a "not unreasonable" value. What is needed are plots showing 'truth' (= known property of training dataset) on x-axis vs 'prediction' (= recovered using AI) on y axis; I would focus on just a few properties/state to keep the ms to a reasonable size, say: $\lambda$, $\chi$, $G$, $\psi$. Such plots will allow a reader to quickly assimilate how well the procedures presented in the paper work.*

AR:  We agree that Section 6 was not properly discussing the AI results. To account for that, we included subsection 6.1, with several plots and discussions. We believe that this addition was pivotal to enhance the quality of the paper.

Sincerely,

Luan Carlos de Sena Monteiro Ozelim, D.Sc.
Corresponding author on behalf of all authors